# Immunological and prognostic significance of novel ferroptosis-related genes in soft tissue sarcoma

**Jiazheng Zhao[1], Yi Zhao[1], Xiaowei Ma[1], Helin Feng[1], Rongmin Cui[2]***

**1** Department of Orthopedics, The Fourth Hospital of Hebei Medical University, Shijiazhuang, Hebei, P. R. China, **2** Department of Operating Room, The Fourth Hospital of Hebei Medical University, Shijiazhuang, Hebei, P. R. China

* rongmincui0311@163.com

**Data Availability Statement:** All data files are available from the GEO database (GSE21122, GSE6481, GSE2719, GSE63157) and TCGA database (TCGA-SARC).

## Abstract

### Background

Ferroptosis has exhibited great potential in the treatment of cancer and has gained widespread attention in soft tissue sarcoma (STS). The aim was to explore the immunological and prognostic significance of novel ferroptosis-related genes in STS.

### Methods

We identified ferroptosis-related differentially expressed genes (DEGs) in STS to construct the networks of enrichment analysis and protein-protein interaction. Subsequently, hub genes with prognostic significance were localized and a series of prognostic and immune analyses were performed.

### Results

40 ferroptosis-related DEGs were identified, of which HELLS, STMN1 EPAS1, CXCL2, NQO1, and IL6 were classified as hub genes and were associated with the prognosis in STS patients. In the results of the immune analysis, PDCD1, CTLA4, TIGIT, IDO1 and CD27 exhibited consistent intense correlations as immune checkpoint genes, as well as macrophage, neutrophil, cytotoxic cell, dendritic cell, interdigitating dendritic cell and plasmacytoid dendritic cell as immune cells. EPAS1 and HELLS might be independent prognostic factors for STS patients, and separate prognostic models were constructed by using them.

### Conclusions

We recognized novel ferroptosis-related genes with prognostic value in STS. Furthermore, we searched out potential immune checkpoints and critical immune cells.

**Funding:** This work was supported by grants from the Project of the Natural Science Foundation of Hebei Province [H2019206309] for HF.The funders had no role in study design, data collection and analysis, decision to publish, or preparation of the manuscript.

**Competing interests:** The authors have declared that no competing interests exist.

## Introduction

Soft tissue sarcoma (STS) is a group of malignant tumors originating from mesenchymal tissue and containing multiple histological subtypes [1]. The prognosis of partial STS is poor with no effective treatment and the precise prediction of the prognosis for STS patients is a challenging topic [2]. The previous view was that immunotherapy was unpromising in STS, but this has been reversed in recent years [3].

Ferroptosis is an emerging phenotype of regulated cell death (RCD) which relies on reactive oxygen species deposition mediated by iron catalysis and lipid peroxidation [4]. Ferroptosis performs an essential role in the initiation, progression and prognosis of multiple diseases [5]. Meanwhile, ferroptosis has exhibited great potential in the treatment of cancer and has gained widespread attention in STS as well [6]. Recent studies have revealed that ferroptosis and tumor immunity can be mutually regulated [7, 8].

In the present study, differentially expressed genes (DEGs) were identified through the Gene Expression Omnibus (GEO) database, the FerrDb database, the Immunology Database and Analysis Portal (ImmPort) database, and the networks of enrichment analysis and protein-protein interaction (PPI) were constructed. Prognostic and immune analyses were performed through the Cancer Genome Atlas (TCGA) database. The aim was to explore the immunological and prognostic significance of novel ferroptosis-related genes in STS.

## Materials and methods

### Data sources

We downloaded RNA-seq data from the GEO database (https://www.ncbi.nlm.nih.gov/geo/) in the GSE21122, GSE6481 and GSE2719 datasets, and all three datasets were from the GPL96 platform. Selected samples from GSE21122 included leiomyosarcoma (26), dedifferentiated liposarcoma(46), myxoid liposarcoma (20), pleomorphic liposarcoma (23), myxofibrosarcoma (31), pleomorphic fibrosarcoma (3), normal human fat (9); selected samples from GSE6481 included synovial sarcoma (16), malignant peripheral nerve sheath tumor (3); selected samples from GSE2719 included gastrointestinal stromal tumor (2), round cell tumor (4). In total, from the GSE21122, GSE6481 and GSE2719 datasets, we selected 174 STS samples covering 10 subtypes as the experimental group and 9 normal human fat samples as the control group for difference analysis. Furthermore, we chose the GSE63157 dataset for external validation of the gene prognostic value. We downloaded RNA-seq and clinical data from the TCGA database (https://www.cancer.gov/about-nci/organization/ccg/research/structural-genomics/tcga) for 263 samples, including leiomyosarcoma (105), dedifferentiated liposarcoma (59), undifferentiated pleomorphic sarcoma (51), myxofibrosarcoma (25), synovial sarcoma (10), malignant peripheral nerve sheath tumor (9), desmoid tumor (2), unclassified sarcoma (2). RNA-seq data in FPKM format was converted to TPM format and log2 transformed. We downloaded the lists of 259 ferroptosis-related genes and 2498 immune-related genes from the FerrDb database (http://www.zhounan.org/ferrdb) [9] and ImmPort database (https://immport.niaid.nih.gov) [10], respectively. All material was sourced from public databases and did not involve informed consent from participants.

### Data pre-processing and differential analysis

We downloaded the GSE2719, GSE6481, and GSE21122 datasets by the GEOquery package of R [11]. Probes with one probe corresponding to more than one molecule were removed, when probes corresponding to the same molecule were encountered, and only the probe with the highest signal value was retained. For the filtered data, we used the ComBat function of the sva

package to remove inter-batch differences, box plots to present the normalization result, principal component analysis (PCA) and uniform manifold approximation and projection (UMAP) plots to present the clustering result (S1A–S1F Fig). Differential analysis was carried out by the limma package [12] and visualized using the ggplot2 package and ComplexHeatmap package [13]. The adjusted $p$ value (false discovery rate, FDR) < 0.05 and | log fold change (FC)| > 1 for the DEGs were set as screening conditions.

## Functional enrichment analysis and PPI networks construction

Gene Ontology (GO) enrichment analysis, Kyoto Encyclopedia of Genes and Genomes (KEGG) enrichment analysis and Gene Set Enrichment Analysis (GSEA) were implemented through the clusterProfiler package of R [14]. FDR < 0.05 for the enriched item was considered statistically significant. After predicting the interactions between DEGs in the Search Tool for the Retrieval of Interacting Genes/Proteins (STRING) database (https://string-db.org/) [15] by setting the combined score > 0.4, the PPI networks were built using Cytoscape [16] and cytoHubba [17] respectively.

## Hub genes identification and prognostic models construction

Through the TCGA database, high expression and low expression groups were divided by the median of DEGs expression and survival analysis was performed with the survival package of R [18]. DEGs with potential prognostic significance were identified as hub genes by log-rank analysis and visualization was achieved through the survminer package. The Wilcoxon rank sum test was chosen for correlation analysis of hub genes expression with clinical variables, and the ggplot2 package was used for visualization. All clinical variables of STS were integrated into univariate Cox regression, parameters were included in overall survival (OS) and progression free survival (PFS), and variables that were significant for univariate analysis were integrated into multivariate Cox regression. After evaluating significant variables in the multivariate analysis by the timeROC package, they were incorporated into a nomogram to construct the model [19]. The population of the model was 263 patients with well-defined STS, from the TCGA database and screened with corresponding clinical information, and the model was validated by a calibration curve, with visualization implemented through the rms package. The results were considered statistically significant at $p < 0.05$.

## Immune analysis

Through the TCGA database, high expression and low expression groups were classified according to the upper and lower quartiles of DEGs expression and the GSVA package of R accompanied by Spearman correlation test was applied for immune analysis [20]. 7 popular immune checkpoint genes (ICGs) [21, 22] and 24 immune cells composing the main tumor immune microenvironment [23] were included by applying the CIBERSORT deconvolution algorithm, and the ggplot2 package was used to construct co-expression plots. The results were considered statistically significant at $p < 0.05$.

## Statistical analysis

Statistical analysis relied on R software (version 3.6.3) and Cytoscape software (version 3.8.2).

## Results

### Ferroptosis-related DEGs identification in STS

A total of 927 DEGs for STS were identified in 183 samples from the GSE21122, GSE6481 and GSE2719 datasets, including 345 for up-regulation and 582 for down-regulation. The volcano plot covered all genes in differential analysis (Fig 1A) and the heatmap displayed the top 20 DEGs for each of up-regulation and down-regulation (Fig 1B). Among them, a total of 40 genes were associated with ferroptosis (Table 1), including 6 for up-regulation and 34 for down-regulation (Fig 1C).

### Ferroptosis-related DEGs enrichment analysis

After conducting enrichment analysis on the 40 ferroptosis-related DEGs, the top 5 enriched entries and pathways were obtained to construct the GO enrichment network (Fig 2A) and the KEGG enrichment network (Fig 2B) respectively. GO analysis indicated that these genes functioned in response to metalion (GO: 0010038), response to corticosteroid (GO: 0031960), response to nutrient levels (GO: 0031667), response to oxidative stress (GO: 0006979) and reactive oxygen species metabolic process (GO: 0072593). KEGG analysis suggested that

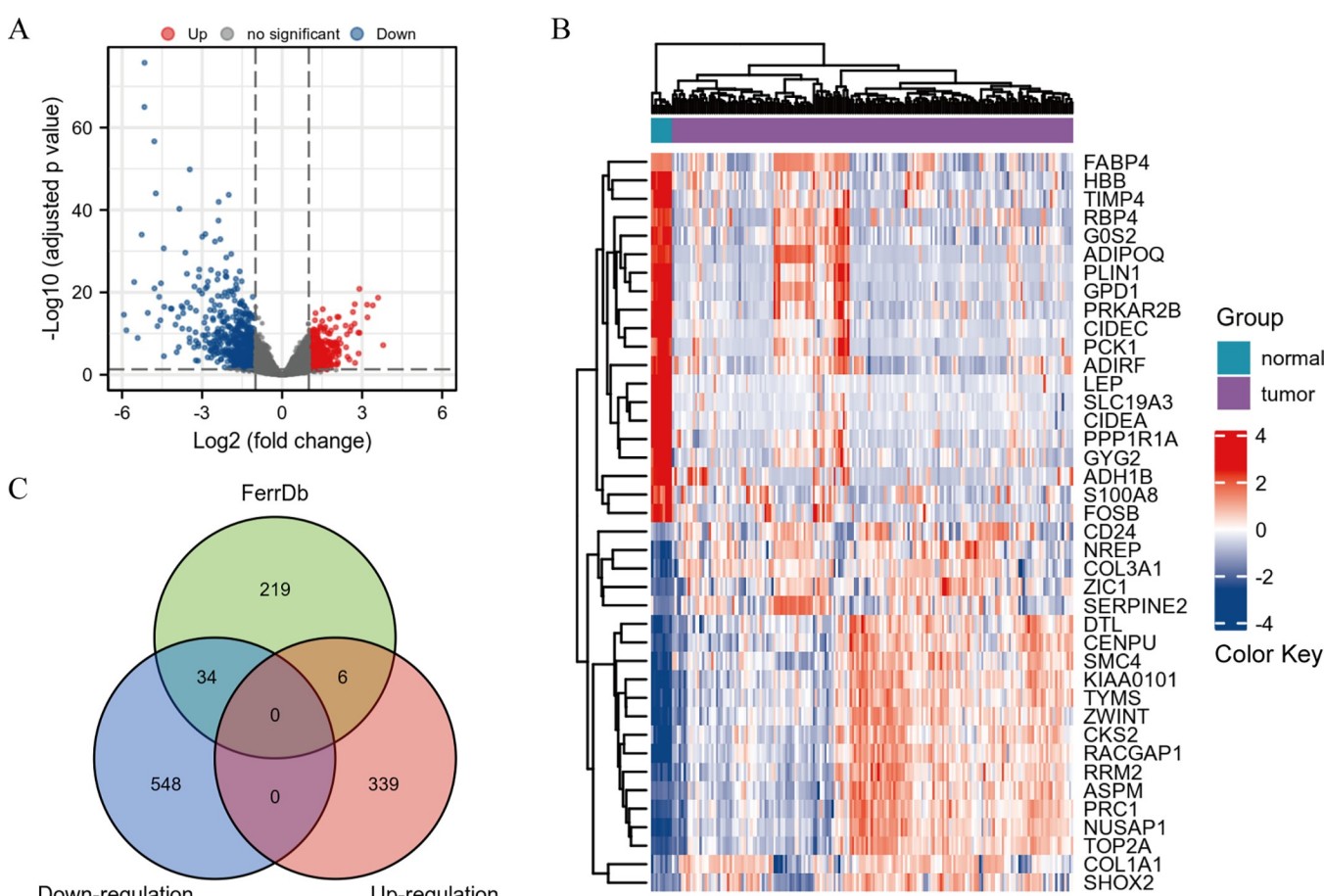

**Fig 1. Identification of ferroptosis-related DEGs in STS.** (A) The volcano plot of all genes. (B) The heatmap of the top 20 DEGs for each of up-regulation and down-regulation. (C) The Venn diagram of the intersection among up-regulated DEGs, down-regulated DEGs and ferroptosis-related genes.

**Table 1. Ferroptosis-related DEGs in STS.**

| No. | Genes | Expression | FDR | log FC |
|---|---|---|---|---|
| 1 | CDKN2A | Up-regulation | <0.001 | 2.012 |
| 2 | HELLS | Up-regulation | <0.001 | 2.015 |
| 3 | GDF15 | Up-regulation | 0.020 | 1.447 |
| 4 | STMN1 | Up-regulation | <0.001 | 1.162 |
| 5 | RRM2 | Up-regulation | <0.001 | 2.550 |
| 6 | AURKA | Up-regulation | <0.001 | 1.447 |
| 7 | PGD | Down-regulation | 0.001 | -1.294 |
| 8 | ACO1 | Down-regulation | <0.001 | -1.714 |
| 9 | GABARAPL1 | Down-regulation | <0.001 | -1.521 |
| 10 | EGFR | Down-regulation | 0.002 | -1.189 |
| 11 | CDO1 | Down-regulation | <0.001 | -2.184 |
| 12 | EPAS1 | Down-regulation | <0.001 | -1.969 |
| 13 | HILPDA | Down-regulation | 0.007 | -1.002 |
| 14 | LPIN1 | Down-regulation | <0.001 | -1.003 |
| 15 | TLR4 | Down-regulation | <0.001 | -1.037 |
| 16 | AKR1C1 | Down-regulation | <0.001 | -2.875 |
| 17 | AKR1C3 | Down-regulation | <0.001 | -2.294 |
| 18 | GCLC | Down-regulation | <0.001 | -1.694 |
| 19 | NQO1 | Down-regulation | <0.001 | -2.045 |
| 20 | MT1G | Down-regulation | <0.001 | -1.565 |
| 21 | SCD | Down-regulation | <0.001 | -2.246 |
| 22 | CDKN1A | Down-regulation | <0.001 | -1.196 |
| 23 | PRDX6 | Down-regulation | <0.001 | -1.083 |
| 24 | PLIN2 | Down-regulation | 0.018 | -1.054 |
| 25 | ZFP36 | Down-regulation | <0.001 | -2.141 |
| 26 | CAV1 | Down-regulation | <0.001 | -1.255 |
| 27 | PTGS2 | Down-regulation | <0.001 | -1.866 |
| 28 | DUSP1 | Down-regulation | <0.001 | -1.566 |
| 29 | NCF2 | Down-regulation | 0.002 | -1.049 |
| 30 | BNIP3 | Down-regulation | 0.001 | -1.602 |
| 31 | PCK2 | Down-regulation | <0.001 | -1.341 |
| 32 | TXNIP | Down-regulation | <0.001 | -1.333 |
| 33 | IL6 | Down-regulation | <0.001 | -2.806 |
| 34 | CXCL2 | Down-regulation | <0.001 | -3.999 |
| 35 | MAP3K5 | Down-regulation | <0.001 | -1.599 |
| 36 | SLC2A3 | Down-regulation | 0.008 | -1.129 |
| 37 | ACSF2 | Down-regulation | <0.001 | -1.181 |
| 38 | TF | Down-regulation | <0.001 | -3.001 |
| 39 | ATF3 | Down-regulation | <0.001 | -1.931 |
| 40 | GPX4 | Down-regulation | <0.001 | -1.325 |

DEGs, differentially expressed genes; STS, soft tissue sarcoma; FC, fold change.

corresponding genes were significantly associated with glutathione metabolism (hsa00480), FoxO signaling pathway (hsa04068), HIF-1 signaling pathway (hsa04066), legionellosis (hsa05134) and Kaposi sarcoma-associated herpesvirus infection (hsa05167).

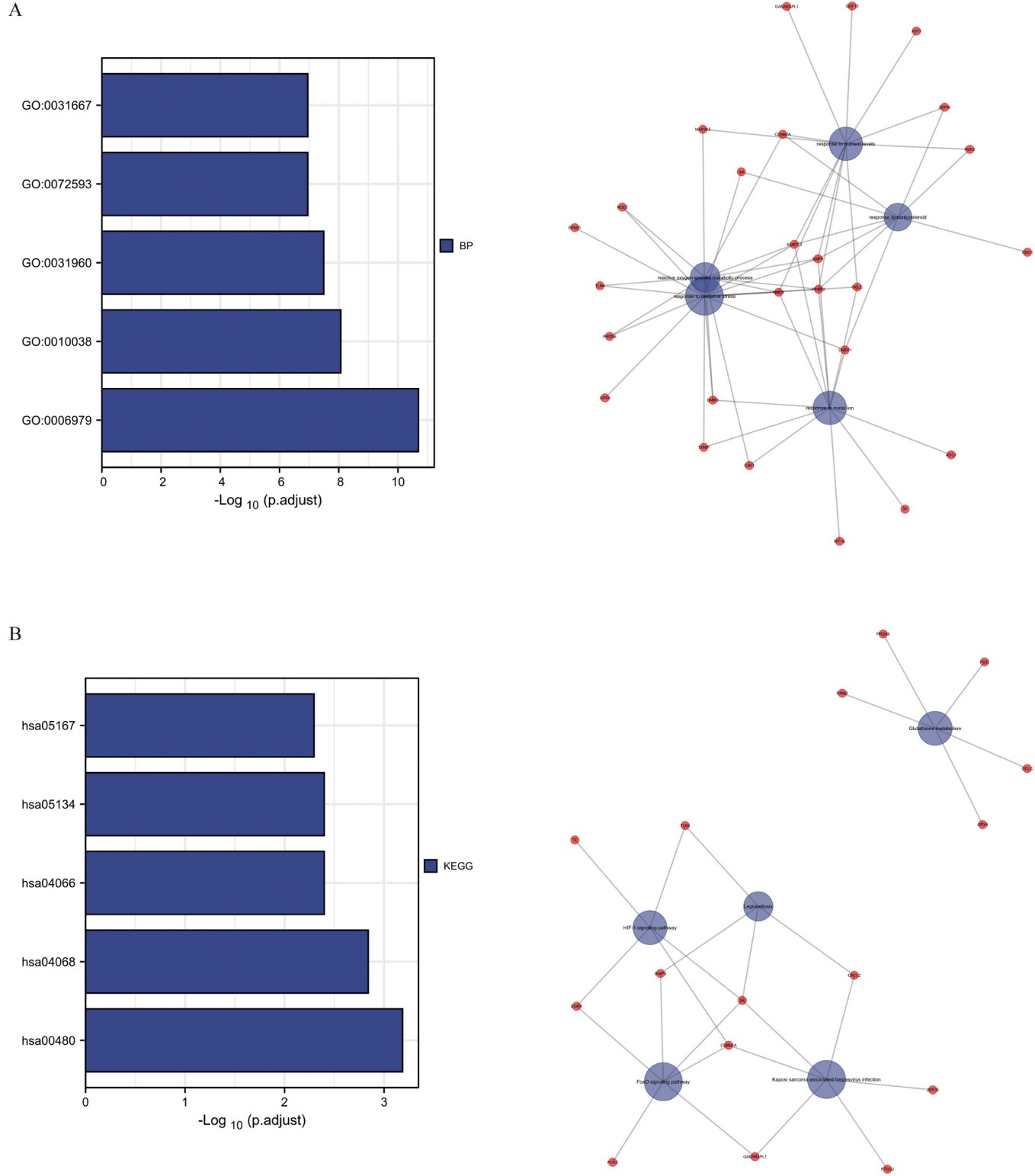

**Fig 2. Enrichment analysis of ferroptosis-related DEGs in STS.** (A) The network of GO enrichment analysis for the top 5 entries. (B) The network of KEGG enrichment analysis for the top 5 pathways.

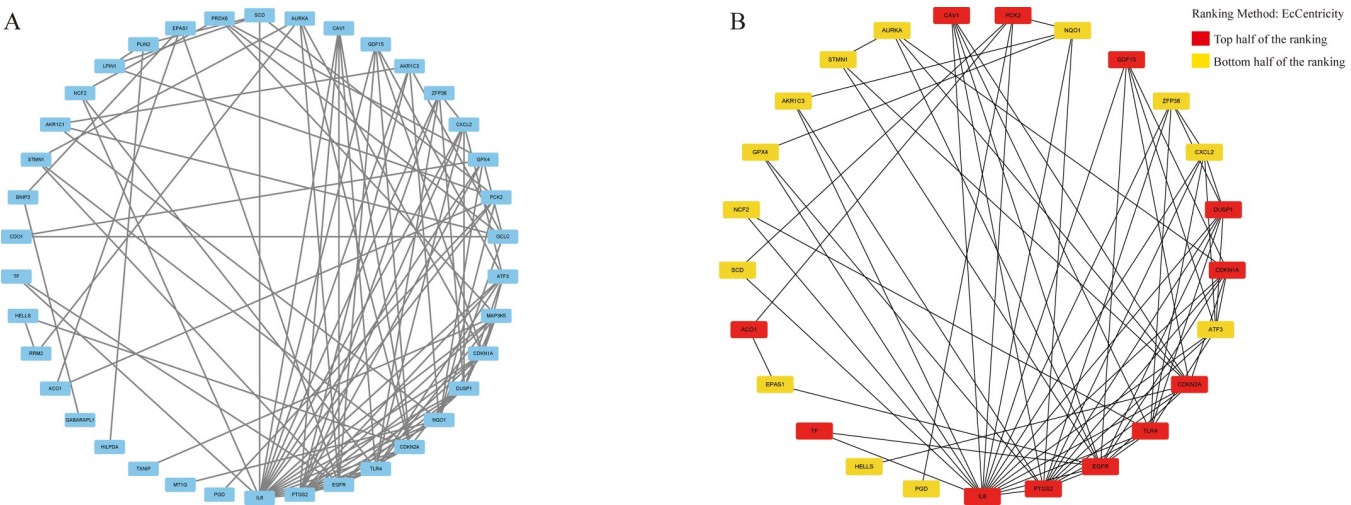

**Fig 3. PPI networks of ferroptosis-related DEGs in STS.** (A) The PPI network covering 38 nodes and 97 edges using Cytoscape. (B) The PPI network covering 25 nodes and 69 edges using cytoHubba.

## PPI networks construction

Interactions of ferroptosis-related DEGs in STS were predicted by STRING and a PPI network covering 38 nodes and 97 edges was structured by Cytoscape (Fig 3A). Subsequently we used cytoHubba to further identify the top 25 genes and build a 25-node, 69-edge PPI network (Fig 3B).

## Hub genes identification

Survival analysis revealed the potential prognostic value of HELLS, STMN1 in up-regulation DEGs and EPAS1, CXCL2, NQO1, IL6 in down-regulation DEGs, with high expression of HELLS, STMN1 and low expression of EPAS1, CXCL2, NQO1, IL6 suggesting a short OS in STS patients (Fig 4A). Accordingly, these 6 genes were identified as hub genes for further

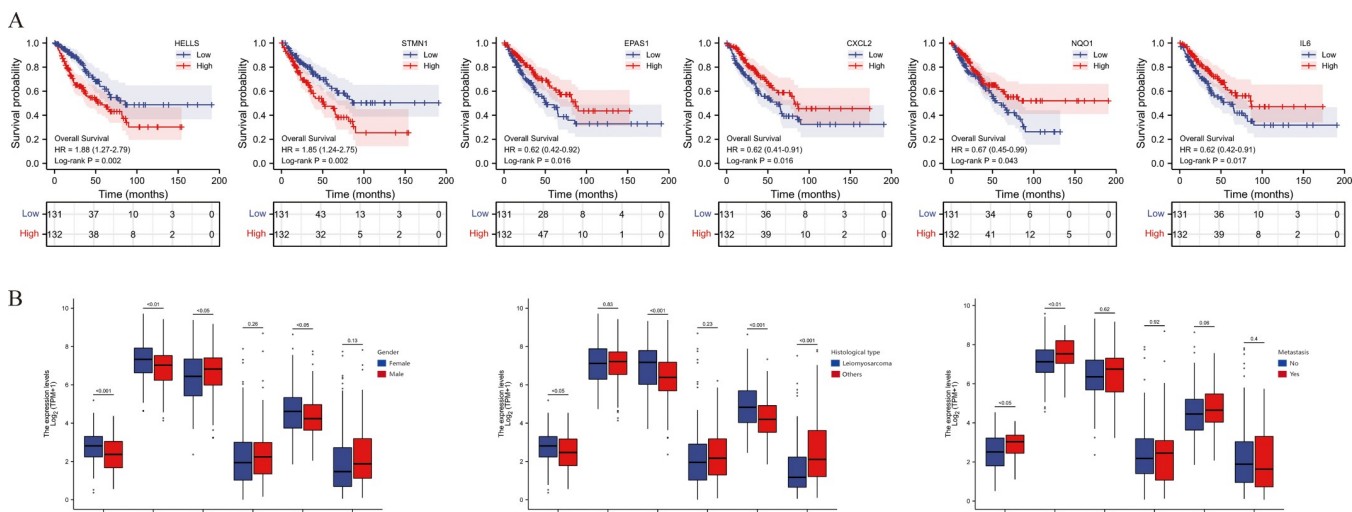

**Fig 4. Clinical relevance of hub genes.** (A) K-M curves of hub genes expression. (B) Box plots of hub genes expression and clinical variables.

study and the association between hub genes and STS clinical variables was analyzed (Fig 4B). Compared to male patients with STS, female patients exhibited high expression of HELLS, STMN1, NQO1 and low expression of EPAS1. Compared to other histological types, leiomyosarcoma showed high expression of HELLS, EPAS1, NQO1 and low expression of IL6. Besides, HELLS and STMN1 was highly expressed in STS metastatic patients compared to non-metastatic patients.

## Hub genes GSEA analysis

The 263 STS samples from TCGA database were divided into low expression and high expression groups based on the median of hub gene expression respectively for GSEA analysis. GSEA manifested significant differences in enrichment of MSigDB Collection (FDR < 0.05) and significant-enriched gene sets were ranked based on normalized enrichment score (NES) values. The top-two most significant-enriched gene sets for HELLS were G alpha signaling events and olfactory transduction (Fig 5A). The top-two most significant-enriched gene sets for STMN1 were signaling by Rho GTPases and processing of capped intron-containing premRNA (Fig 5B). The top-two most significant-enriched gene sets for EPAS1 were M-phase and metabolism of amino acids and derivatives (Fig 5C). The top-two most significant-enriched gene sets for CXCL2 were neuronal system and neuroactive ligand receptor interaction (Fig 5D). The top-two most significant-enriched gene sets for NQO1 were signaling by interleukins and Leishmania infection (Fig 5E). The top-two most significant-enriched gene sets for IL6 were signaling by interleukins and GPCR-ligand binding (Fig 5F).

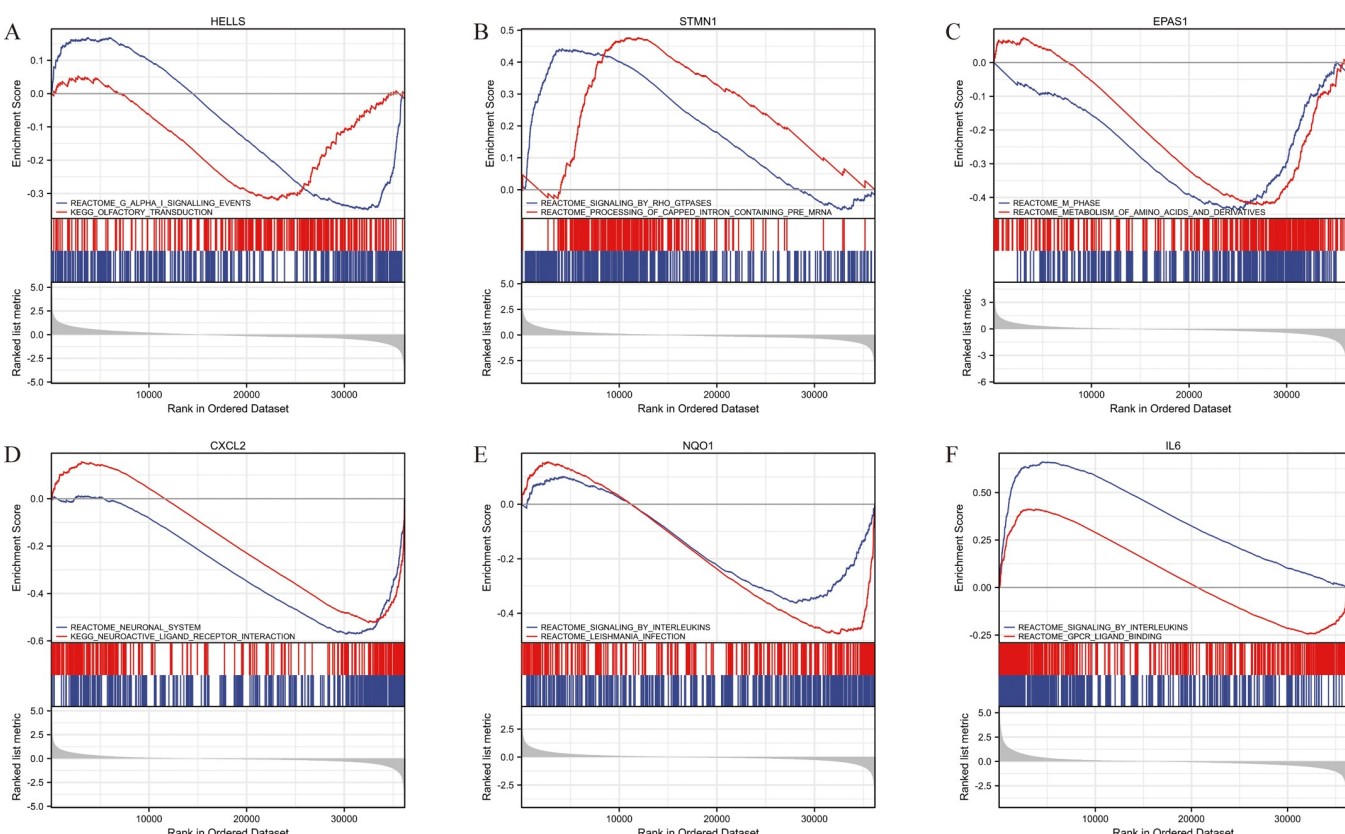

**Fig 5. GSEA analysis of hub genes.** (A) HELLS. (B) STMN1. (C) EPAS1. (D) CXCL2. (E) NQO1. (F) IL6.

## EPAS1 and HELLS might be independent prognostic factors for STS patients

Clinical variables for STS and the expression of 6 hub genes were included in univariate Cox regression analysis and those factors of significance were further subsumed into multivariate Cox regression analysis. The results revealed that when the prognostic indicator was OS, high grade residual tumor, metastasis, positive margin status, high expression of HELLS and STMN1, low expression of EPAS1, CXCL2, NQO1, IL6 were associated with poor prognosis. Furthermore, residual tumor, metastasis status, margin status, EPAS1 expression might be independent prognostic factors for OS in STS patients (Table 2). When the prognostic indicator was PFS, high grade residual tumor, metastasis, positive margin status, HELLS high expression were associated with poor prognosis. Residual tumor, metastasis status, margin status, HELLS expression might be independent prognostic factors for PFS in STS patients (Table 3).

## Validation of EPAS1 and HELLS prognostic value

Predictive efficacy of EPAS1 and HELLS for prognosis was internally verified using time-dependent receiver operating characteristic (ROC) curves in TCGA database (Fig 6A and 6B). Subsequently, predictive efficacy of EPAS1 and HELLS for prognosis was externally validated using time-dependent ROC curves in GEO database, which exhibited similar prognostic value (Fig 6C and 6D).

## Construction and evaluation of prognostic models for STS patients

The statistically significant results of the multivariate Cox regression analysis were used to construct the separate nomogram for prediction models of OS (Fig 7A) and PFS (Fig 7B) in STS patients. For both patients with primary STS and metastatic STS, the indicators for each nomogram were derived from the primary tumor foci. The C-indexes for OS and PFS model were 0.756 (0.719–0.794) and 0.782 (0.756–0.808) respectively. Calibration curves for the

**Table 2. Univariate and multivariate Cox regression analysis to identify prognostic factors for OS in patients with STS.**

| Variables | Total(N) | Univariate analysis | | Multivariate analysis | |
|---|---|---|---|---|---|
| | | Hazard ratio (95% CI) | *p*-value | Hazard ratio (95% CI) | *p*-value |
| Age (>60 vs. < = 60) | 263 | 1.285 (0.864–1.911) | 0.216 | | |
| Gender (Male vs. Female) | 263 | 0.905 (0.607–1.349) | 0.623 | | |
| Race (White vs. Others) | 254 | 0.725 (0.350–1.501) | 0.386 | | |
| Histological type (Leiomyosarcoma vs. Others) | 263 | 0.913 (0.611–1.363) | 0.656 | | |
| Radiation therapy (Yes vs. No) | 257 | 0.864 (0.557–1.339) | 0.513 | | |
| Residual tumor (R2 vs. R0&R1) | 235 | 8.365 (3.972–17.617) | <0.001 | 22.480 (6.480–77.987) | <0.001 |
| Metastasis status (Yes vs. No) | 179 | 2.888 (1.762–4.732) | <0.001 | 3.493 (1.852–6.585) | <0.001 |
| Margin status (Positive vs. Negative) | 213 | 1.957 (1.215–3.151) | 0.006 | 1.879 (1.054–3.350) | 0.032 |
| HELLS (High vs. Low) | 263 | 1.883 (1.250–2.836) | 0.002 | 1.272 (0.603–2.683) | 0.527 |
| STMN1 (High vs. Low) | 263 | 1.859 (1.242–2.783) | 0.003 | 0.844 (0.416–1.712) | 0.638 |
| EPAS1 (Low vs. High) | 263 | 1.627 (1.093–2.424) | 0.017 | 2.698 (1.347–5.406) | 0.005 |
| CXCL2 (Low vs. High) | 263 | 1.625 (1.089–2.425) | 0.017 | 1.142 (0.531–2.455) | 0.735 |
| NQO1 (Low vs. High) | 263 | 1.504 (1.009–2.242) | 0.045 | 1.307 (0.737–2.319) | 0.360 |
| IL6 (Low vs. High) | 263 | 1.624 (1.085–2.432) | 0.018 | 1.191 (0.557–2.544) | 0.652 |

OS, overall survival; STS, soft tissue sarcoma.

**Table 3. Univariate and multivariate Cox regression analysis to identify prognostic factors for PFS in patients with STS.**

| Variables | Total(N) | Univariate analysis | | Multivariate analysis | |
|---|---|---|---|---|---|
| | | Hazard ratio (95% CI) | p-value | Hazard ratio (95% CI) | p-value |
| Age (>60 vs. < = 60) | 263 | 0.938 (0.675–1.305) | 0.706 | | |
| Gender (Male vs. Female) | 263 | 1.092 (0.785–1.520) | 0.600 | | |
| Race (White vs. Others) | 254 | 1.155 (0.605–2.203) | 0.662 | | |
| Histological type (Leiomyosarcoma vs. Others) | 263 | 1.101 (0.790–1.536) | 0.570 | | |
| Radiation therapy (Yes vs. No) | 257 | 1.124 (0.788–1.602) | 0.519 | | |
| Residual tumor (R2 vs. R0&R1) | 235 | 4.230 (2.140–8.360) | <0.001 | 4.985 (1.811–13.723) | 0.002 |
| Metastasis status (Yes vs. No) | 179 | 7.294 (4.700–11.318) | <0.001 | 6.672 (4.087–10.894) | <0.001 |
| Margin status (Positive vs. Negative) | 213 | 2.176 (1.493–3.173) | <0.001 | 2.497 (1.551–4.021) | <0.001 |
| HELLS (High vs. Low) | 263 | 1.549 (1.111–2.160) | 0.010 | 1.707 (1.040–2.803) | 0.035 |
| STMN1 (High vs. Low) | 263 | 1.368 (0.981–1.908) | 0.064 | | |
| EPAS1 (Low vs. High) | 263 | 1.053 (0.757–1.464) | 0.760 | | |
| CXCL2 (Low vs. High) | 263 | 1.017 (0.731–1.413) | 0.922 | | |
| NQO1 (Low vs. High) | 263 | 1.343 (0.965–1.870) | 0.081 | | |
| IL6 (Low vs. High) | 263 | 1.303 (0.937–1.813) | 0.116 | | |

PFS, progression free survival; STS, soft tissue sarcoma.

models of OS (Fig 7C) and PFS (Fig 7D) confirmed the consistency of the predicted prognosis with the actual outcome.

## Association of hub genes expression and ICGs

CXCL2 and IL6 were shown to be immunologically relevant in 6 hub genes (Fig 8A). We correlated hub genes with ICGs and presented the results in co-expression heatmaps. CXCL2 and IL6 showed consistent results, with both CXCL2 and IL6 positively linked to the expression of PDCD1, CTLA4, TIGIT, IDO1 and CD27 (Fig 8B and 8C). Consistency of results and significant association with ICGs were not demonstrated in HELLS, STMN1, EPAS1 and NQO1 (Fig 8D–8G).

## Association of hub genes expression and immune cells infiltration

6 hub genes were subsequently correlated with 24 immune cells in the tumor microenvironment (Fig 9A). In addition to CXCL2 and IL6, we observed that HELLS was also strongly associated with immune cells and exhibited the consistent result with CXCL2 and IL6. CXCL2 and IL6 with down-regulated in STS were significantly positively related to macrophage, neutrophil, cytotoxic cell, dendritic cell (DC), interdigitating dendritic cell (iDC), plasmacytoid dendritic cell (pDC) (all r > 0.3) (Fig 9B and 9C), and HELLS with up-regulated in STS was comparatively negatively correlated with macrophage, neutrophil, cytotoxic cell, DC, iDC, pDC (all r < -0.3) (Fig 9D) Consistency of results and significant association with immune cells were not demonstrated in STMN1, EPAS1 and NQO1.

## Discussion

STS is a set of heterogeneous malignancies involving over 100 different histological types, with widely varying treatment outcomes [24]. In general, current therapies are only effective in a small proportion of STS, with limited efficacy in most STS and even recurrence in more than 50% of patients [25]. Although it was considered that STS was extremely insensitive to

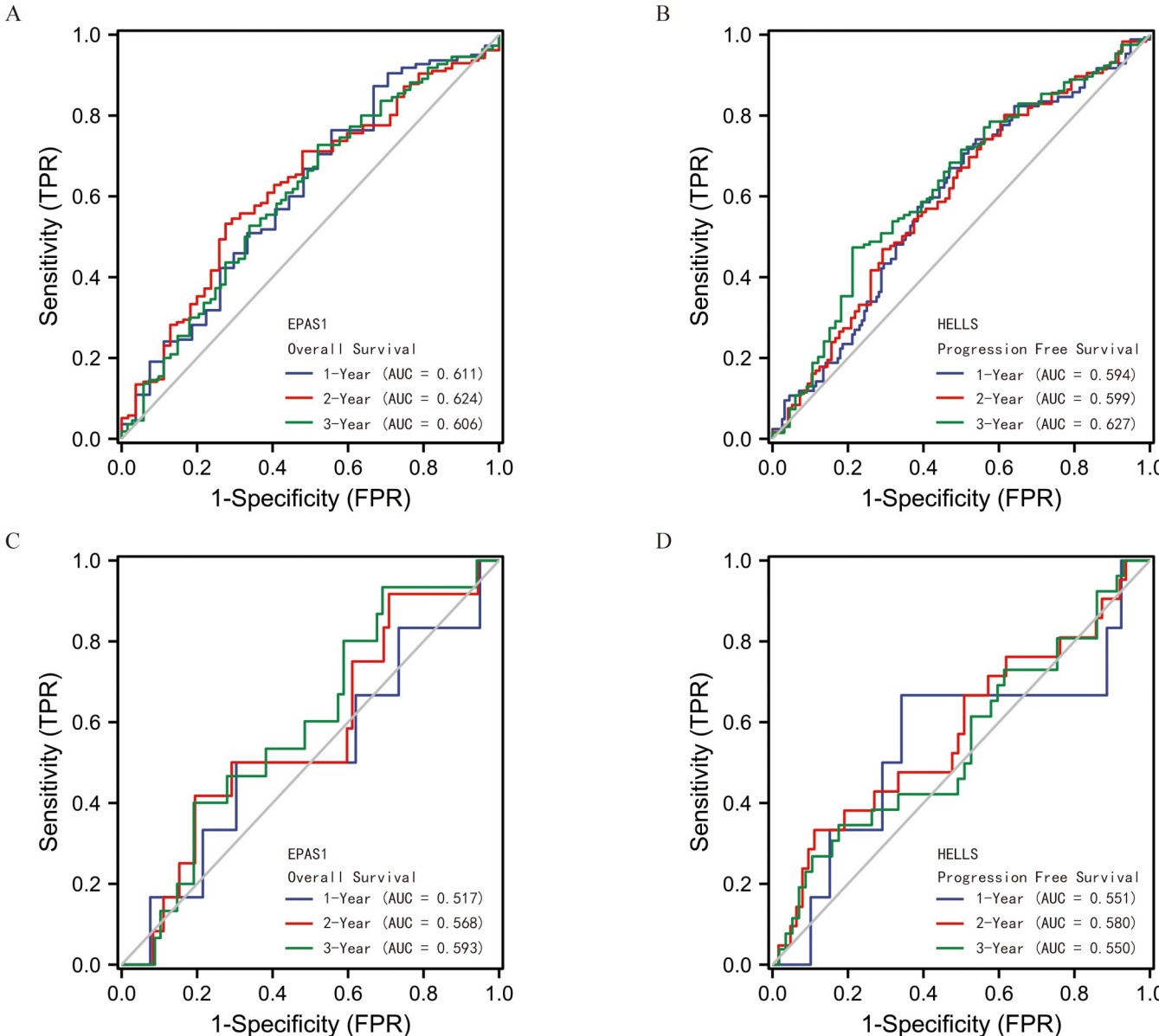

**Fig 6. Validation of EPAS1 and HELLS prognostic value.** (A) The ROC curve of EPAS1 predicting OS for STS patients in TCGA database. (B) The ROC curve of HELLS predicting PFS for STS patients in TCGA database. (C) The ROC curve of EPAS1 predicting OS for STS patients in GEO database. (D) The ROC curve of HELLS predicting PFS for STS patients in GEO database.

immune responses in the past, which precluded the application of immunotherapy to STS, recent studies have demonstrated a large degree of immune heterogeneity within the subclass of STS and some positive responses to immunotherapy have also been reported in successive clinical trials [26, 27]. Partial STS subtypes, including dedifferentiated liposarcoma, leiomyosarcoma, embryonal rhabdomyosarcoma and undifferentiated pleomorphic sarcoma, have been identified as featuring high levels of immune cells infiltration and ICGs expression, and exhibit a potentially active reaction to immune checkpoint inhibitors (ICIs) therapy [3]. Consequently it is essential to locate critical ICGs and immune infiltration factors adapted to STS.

Currently, the availability of immunotherapy alone is severely limited in patients with most tumor types. Since extensive crossover between immunotherapy and non-apoptotic RCD

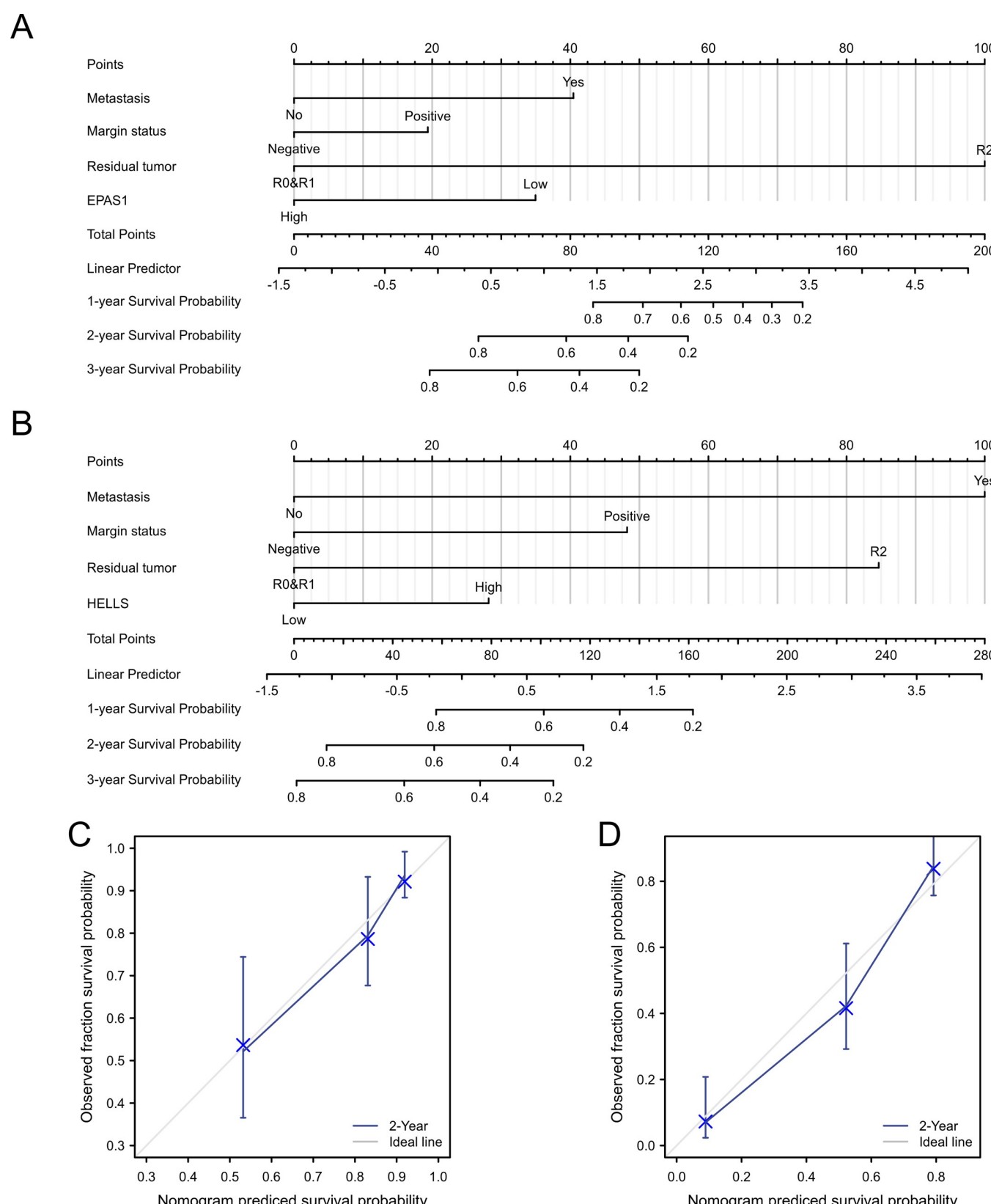

**Fig 7. Visualization of prognostic prediction models in STS.** (A) The nomogram for predicting OS. (B) The nomogram for predicting PFS. (C) The calibration curve to evaluate the OS nomogram. (D) The calibration curve to evaluate the PFS nomogram.

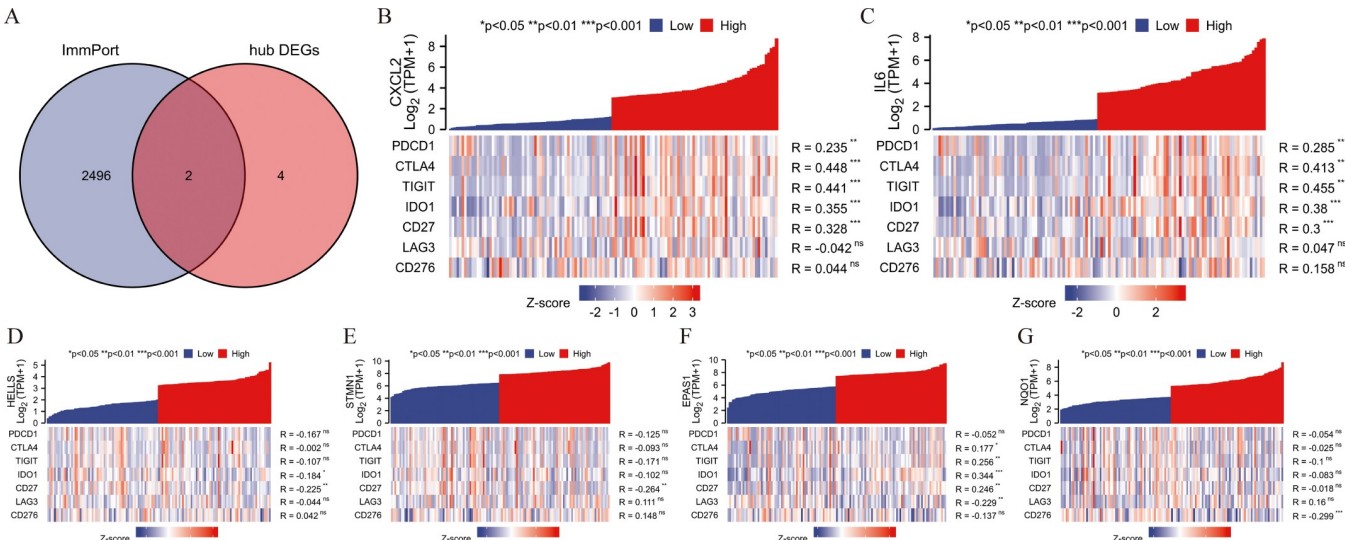

**Fig 8. Analysis of ICGs in STS.** (A) The Venn diagram of the intersection between hub genes and immune-related genes. The correlation of ICGs with the expression of CXCL2 (B), IL6 (C), HELLS (D), STMN1 (E), EPAS1 (F), NQO1 (G).

mechanisms has been detected, non-apoptotic cancer cell death accompanied by immunomodulation is considered an exceedingly promising strategy for cancer treatment [28]. Ferroptosis, a neoteric form of RCD with unique biological and morphological features, has been shown to interact with the tumor immune response and can influence immunotherapeutic efficacy on the one hand [8], and in turn is regulated by immune cells on the other [7]. In the present study, based on ferroptosis-related genes in STS, we identified potential ICGs including PDCD1, CTLA4, TIGIT, IDO1 and CD27, which might serve as important targets for immunotherapy. In addition, we explored a group of closely related immune cells including macrophage, neutrophil, cytotoxic cell, DC, iDC and pDC, which might act as pivotal regulators in the immune microenvironment of STS. Interestingly, we observed high concordance of immune analysis results for HELLS with CXCL2 and IL6, revealing for the first time a possible immunological effect of HELLS in tumor.

Among dedifferentiated liposarcoma, undifferentiated pleomorphic sarcoma and leiomyosarcoma, it has been confirmed that tumors with high immunogenic gene profiles are accompanied by high levels of PDCD1 expression [29]. PD-1, as the most researched immune checkpoint, is encoded by PDCD1 and also occupies an important position in STS study. More than half of the samples in a STS cohort had positive expression of PD-1 on immune cells [30], and PD-1 expression is also generally considered to be associated with the prognosis of STS patients [31, 32]. Moreover, CTLA4, IDO1 and other ICGs have demonstrated varying degrees of value for STS management [3]. In terms of immune cells, macrophage has been established as a significant player in several sarcoma types [33], with the modification of the macrophage phenotype from tumor-promoting to tumor-suppressing regarded as a promising option for STS treatment [34]. And a range of immunotherapies targeting DC, iDC and pDC may be well tolerated in patients with refractory STS due to their excellent immunological response and safety profile, as well as offering the opportunity to prevent recurrence of sarcoma [35]. On balance, for most STS subtypes, immunotherapy may be required novel regimens and combinations [34].

In addition, we substantiated that EPAS1 and HELLS might act as independent prognostic predictors of STS, leading to the construction of two efficient prognostic models. For both

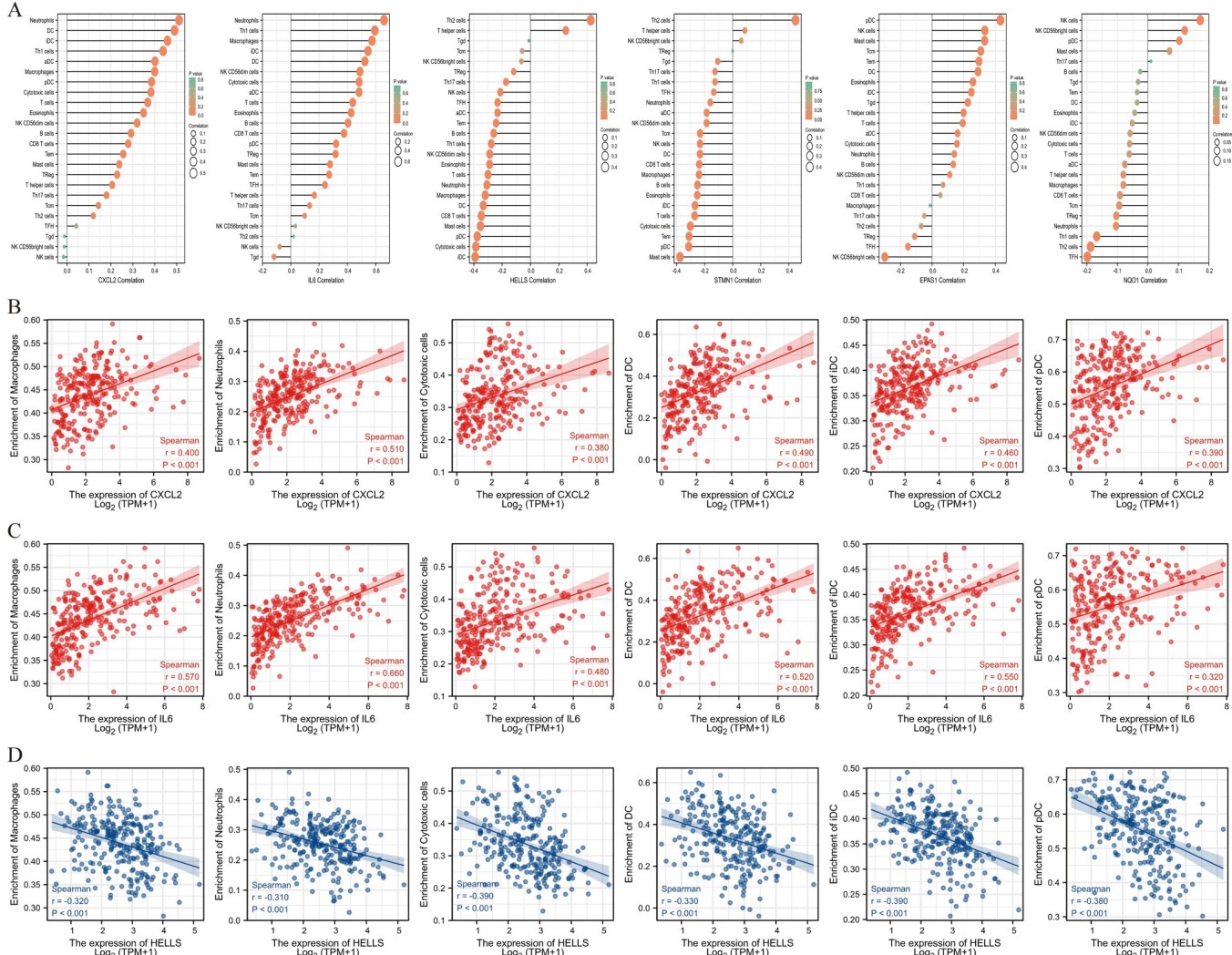

**Fig 9. Analysis of immune cells in STS.** (A) The correlation of 6 hub genes expression with 24 immune cells. (B) The correlation of CXCL2 expression with 6 immune cells. (C) The correlation of IL6 expression with 6 immune cells. (D) The correlation of HELLS expression with 6 immune cells.

patients with primary STS and metastatic STS, the indicators for each nomogram were derived from the primary tumor foci. However, the obtained model still needs to be further verified in an independent cohort. The expression of 6 hub genes was discovered to be associated with survival during the model construction, with EPAS1, STMN1, CXCL2, NQO1 being identified for the first time in STS. EPAS1 is a diver of ferroptosis [36], compared to normal tissue, which is expressed at lower levels in most human STS [37]. Zhu et al. found that up-regulation of EPAS1 significantly enhanced the growth inhibition of gastric adenocarcinoma and that targeting EPAS1 might be an alternative therapeutic approach for cancer [38]. Relatively, HELLS, NQO1 are suppressors of ferroptosis [39, 40]. Law et al. suggested that HELLS mediated epigenetic silencing of various cancer suppressor genes and evidenced in hepatocellular carcinoma that its overexpression potentiated tumor cell migration and proliferation [41]. Huang et al. identified high expression and prognostic impact of HELLS in STS samples [42], which also underpinned our findings. In the TCGA database of STS samples, GSEA indicated that NQO1 was closely connected to interleukin-related signaling pathways. NQO1 has been confirmed to

interact with interleukins in a variety of cancers, thereby affecting the inflammatory response and participating in the immune regulation associated with the tumor microenvironment [43, 44]. As for STMN1, CXCL2 and IL6, they are currently treated as biomarkers of ferroptosis and their expression is monitored for down-regulation once ferroptosis occurs [45, 46]. STMN1 is commonly recognized as an oncogene, and its up-regulation is tightly linked to the malignant behaviour and poor prognosis of various tumors [47]. In leiomyosarcoma, STMN1 has also been characterised by high expression and can be a sensitive biomarker with strong diagnostic efficacy [48, 49]. Our study revealed the potential immunological relevance and clinical value of these novel ferroptosis-related genes, which might contribute to the precise treatment and prognostic prediction of patients with STS.

## Conclusions

In conclusion, we identified novel ferroptosis-related genes with prognostic value in STS. Furthermore, we searched out potential immune checkpoints and critical immune cells.

## Supporting information

**S1 Fig. Evaluation of data pre-processing from the GEO database.** Comparison of box plots (A-B), PCA plots (C-D) and UMAP plots (E-F) before and after data pre-processing. (TIF)

## Author Contributions

**Conceptualization:** Jiazheng Zhao, Rongmin Cui.

**Formal analysis:** Jiazheng Zhao, Helin Feng, Rongmin Cui.

**Methodology:** Jiazheng Zhao, Helin Feng, Rongmin Cui.

**Validation:** Yi Zhao, Xiaowei Ma.

**Visualization:** Yi Zhao, Xiaowei Ma.

**Writing – original draft:** Jiazheng Zhao.

**Writing – review & editing:** Helin Feng, Rongmin Cui.

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
