## [Decision Letter · Decision Letter 0]

20 Oct 2021

PONE-D-21-24357Immunological and prognostic significance of novel ferroptosis-related genes in soft tissue sarcomaPLOS ONE

Dear Dr. Feng,

Thank you for submitting your manuscript to PLOS ONE. After careful consideration, we feel that it has merit but does not fully meet PLOS ONE’s publication criteria as it currently stands. Therefore, we invite you to submit a revised version of the manuscript that addresses the points raised during the review process.

We look forward to receiving your revised manuscript.

Kind regards,

Sandro Pasquali, M.D., Ph.D.

Academic Editor

PLOS ONE

Journal Requirements:

Reviewers' comments:

Reviewer's Responses to Questions

**Comments to the Author**

1. Is the manuscript technically sound, and do the data support the conclusions?

Reviewer #1: Partly

Reviewer #2: Yes

2. Has the statistical analysis been performed appropriately and rigorously? 

Reviewer #1: Yes

Reviewer #2: Yes

3. Have the authors made all data underlying the findings in their manuscript fully available?

Reviewer #1: Yes

Reviewer #2: Yes

4. Is the manuscript presented in an intelligible fashion and written in standard English?

Reviewer #1: Yes

Reviewer #2: Yes

5. Review Comments to the Author

Reviewer #1: In this manuscript Zhao et coworkers performed an in silico analysis of RNAseq data from a very large series of well characterized soft tissue sarcoma (STS) samples spanning different subtypes with the aim to explore the immunological and prognostic value of ferroptosis-related genes, in particular of 6 genes (HELLS, STMN1, EAPS1, CXCL2, NQO1, IL6).

The here presented data are interesting, but the manuscript is too descriptive and, in some points, confused. In particular, authors should address the following major issues:

-Authors should clearly specify which samples they compared to identify genes differentially express in STS. Did they compare tumor vs normal tissues?

-Authors should add in the main text a table reporting the name of the 40 ferroptosis-related genes along with their fold change.

-Results reported in figure 4B are not commented in the text. Authors should discuss the differential expression of the 6 hub genes between male and female, leiomyosarcoma and other STS subtypes, different disease progression.

-Authors should comment and discuss gene sets differentially enriched in STS TCGA samples divided on the basis of expression of the 6 hub genes.

-Paragraphs entitled ‘Association of hub genes expression and ICGs’ and ‘Association of hub genes expression and immune cells infiltration’ should be moved at the end of Results section.

-Authors should specify which deconvolution tool they used to estimate population of immune cells infiltrating the tumors.

-‘Immunohistochemical validation’ paragraph should be removed, as in my opinion it does not represent a validation of results obtained by in silico analysis.

-Extensive editing of English language is required.

Minor issues:

-In Figure 2, names of GO and KEGG networks are unreadable. May be authors should add them in the bar graph.

-In Figure 3, color legend is missing. Please, add it.

Reviewer #2: Thank you for putting together this comprehensive manuscript that identifies several novel targets for immunotherapy agents derived from the ferroptosis pathway in sarcoma.

It will be interesting to how you intend to translate these findings into clinical trials.

6. PLOS authors have the option to publish the peer review history of their article (what does this mean?). If published, this will include your full peer review and any attached files.

Reviewer #1: No

Reviewer #2: No

---

## [Author Response · Author response to Decision Letter 0]

1 Nov 2021

Dear editor,

Thank you very much for your letter and advice. We have revised the paper, and would like to re-submit it for your consideration. We have addressed the comments raised by the reviewers, and the amendments are highlighted in red in Revised Manuscript with Track Changes. 

We hope that the revision is acceptable, and we look forward to hearing from you soon. 

With best wishes,

Yours sincerely,

We would like to express our sincere gratitude to the reviewers for their constructive and positive comments. It is an honor for our manuscript to receive such rigorous and meticulous guidance. Explain accordingly one by one:

Reviewer #1 comments:

-Authors should clearly specify which samples they compared to identify genes differentially express in STS. Did they compare tumor vs normal tissues? Thank you for your insightful suggestion. We have added a detailed explanation and red-flagged it in the method section of the article.

-Authors should add in the main text a table reporting the name of the 40 ferroptosis-related genes along with their fold change. This is a great recommendation. We have added Table 1 to show that 40 ferroptosis-related genes.

-Results reported in figure 4B are not commented in the text. Authors should discuss the differential expression of the 6 hub genes between male and female, leiomyosarcoma and other STS subtypes, different disease progression. Thanks for the comment. We have added a detailed explanation of the clinical relevance of the hub gene in the results section of the article and have highlighted it in red.

-Authors should comment and discuss gene sets differentially enriched in STS TCGA samples divided on the basis of expression of the 6 hub genes. Thanks for your reminder. We have added a description of the GSEA results in the results section of the article and discussed the relevant results in the discussion section. All of the above changes are highlighted in red in the article.

-Paragraphs entitled ‘Association of hub genes expression and ICGs’ and ‘Association of hub genes expression and immune cells infiltration’ should be moved at the end of Results section. Thank you for your suggestion. They have been moved at the end of Results section.

-Authors should specify which deconvolution tool they used to estimate population of immune cells infiltrating the tumors. Thanks for your reminder. We have added the relevant information and red-flagged it in the method section of the article.

-‘Immunohistochemical validation’ paragraph should be removed, as in my opinion it does not represent a validation of results obtained by in silico analysis. Thanks for the comment. ’Immunohistochemical validation’ paragraph has been removed.

-Extensive editing of English language is required. Thank you for your insightful suggestion. In response to the language issues in the article, we have carefully reviewed and gone through extensive revisions. Subsequently, we found multiple English language professionals to review and correct the grammar and vocabulary in our article. Thank you again for your reminder.

-In Figure 2, names of GO and KEGG networks are unreadable. May be authors should add them in the bar graph. Thank you for your suggestion. Since the names of both KEGG and GO analysis results are too long, and the size of Figure is limited to the extent that they cannot be shown clearly in the bar chart. Therefore, we chose to provide additional descriptions of the names and numbers of the KEGG and GO analysis results in the Results section of the article, which are highlighted in red.

-In Figure 3, color legend is missing. Please, add it. Thanks for the comment. We have added them in the Figure 3.

Reviewer #2 comments:

Thank you for putting together this comprehensive manuscript that identifies several novel targets for immunotherapy agents derived from the ferroptosis pathway in sarcoma. It will be interesting to how you intend to translate these findings into clinical trials. Thanks for your reminder. This is one of the few comprehensive analyses for soft tissue sarcoma (STS) with large samples, multiple subtypes and multiple databases. Ferroptosis, immune checkpoint, immune cell and prognostic model were integrated. Initially, we identified novel ferroptosis-related genes with prognostic value in STS. Furthermore, we searched out potential key immune checkpoints and immune cells, and revealed for the first time their possible immune relevance. Ultimately, we constructed two efficient predictive models for prognosis of STS patients. 

The results of our current study are a guide to translate and carry out subsequent experiments. The first is to carry out relevant experiments around ferroptosis, and to carry out ROS, GPX4, and Fe level measurements in STS samples around the six ferroptosis-related hub genes we identified, and to verify the relationship with related pathway molecules based on the results of enrichment analyses to find upstream and downstream mechanisms. Second, subsequent experiments can be centered on precise immunotherapy of STS. Since the original view was that immunotherapy for STS was almost useless, immunotherapy for STS is now possible with the discovery of novel RCD mechanisms such as ferroptosis. Due to the heterogeneity of STS, immunotherapy around STS requires a very high degree of precision. It is crucial to clarify the subtypes of STS that are sensitive to immunotherapy and to specify the immune cells, immune checkpoint genes, and reciprocal genes involved in STS immunotherapy. The immune cells and immune checkpoint genes identified in our current study with the broad category of STS are theoretically applicable to immunological studies of all STS subtypes. In addition, we found the immunogenicity of HELLS for the first time, and HELLS plays an important role in immunological studies of STS as a ferroptosis-related gene, which also has an innovative guiding effect on the development of related experiments. The final aspect involved is the validation of the clinical data, and the 2 prognostic models we designed were applicable to cohort studies of all STS subtypes. Of course to the extent that the bioinformatics analysis we performed only served as a guide, the definitive conclusions need to be validated by a large number of follow-up experiments.

---

## [Decision Letter · Decision Letter 1]

15 Nov 2021

PONE-D-21-24357R1Immunological and prognostic significance of novel ferroptosis-related genes in soft tissue sarcomaPLOS ONE

Dear Dr. Cui,

Thank you for submitting your manuscript to PLOS ONE. After careful consideration, we feel that it has merit but does not fully meet PLOS ONE’s publication criteria as it currently stands. Therefore, we invite you to submit a revised version of the manuscript that addresses the points raised during the review process.

Authors did a great job in addressing comments from a reviewer. The manuscript has been furtherly reviewed by a bioinformatician and comments provided. Also, some methodological comments on nomograms need to be addressed. 

We look forward to receiving your revised manuscript.

Kind regards,

Sandro Pasquali, M.D., Ph.D.

Academic Editor

PLOS ONE

Journal Requirements:

Additional Editor Comments:

Regarding the generation of prognostic nomograms, some clarifications and modifications are needed. The authors should consider the following publication "Kattan MW, Hess KR, Amin MB, et al;members of the AJCC Precision MedicineCore. American Joint Committee on Cancer acceptance criteria for inclusion of riskmodels for individualized prognosis in th epractice of precision medicine. CA Cancer J Clin. 2016;66:370–374." as a guidence to develop and present their nomograms. In this article there is checklist, which should be followed. For instance, it is unclear which is the patient population (e.g. primary or metastatic tumour) that was used to generate the nomograms and it is unclear if nomograms were validated. Although I understand that Authors do not aim at targeting inclusion in AJCC staging, if the authors cannot match the required quality standards for their nomograms it would be probably better to exclude them from the current article.

Reviewers' comments:

Reviewer's Responses to Questions

**Comments to the Author**

1. If the authors have adequately addressed your comments raised in a previous round of review and you feel that this manuscript is now acceptable for publication, you may indicate that here to bypass the “Comments to the Author” section, enter your conflict of interest statement in the “Confidential to Editor” section, and submit your "Accept" recommendation.

Reviewer #1: All comments have been addressed

Reviewer #3: (No Response)

2. Is the manuscript technically sound, and do the data support the conclusions?

Reviewer #1: (No Response)

Reviewer #3: Yes

3. Has the statistical analysis been performed appropriately and rigorously? 

Reviewer #1: (No Response)

Reviewer #3: Yes

4. Have the authors made all data underlying the findings in their manuscript fully available?

Reviewer #1: (No Response)

Reviewer #3: Yes

5. Is the manuscript presented in an intelligible fashion and written in standard English?

Reviewer #1: (No Response)

Reviewer #3: Yes

6. Review Comments to the Author

Reviewer #1: The authors have addressed all my questions and made the necessary changes to the manuscript. I have no further comments.

Reviewer #3: Authors describe a panel of genes, related to ferroptosis able to have a significative discrimination power in prognostic and immunological model of soft tissue sarcomas.

This article has an innovative character and could contribute to knowledge of soft tissue sarcoma.

Minor reviews:

1) Authors should specify which metric they used for ranking genes in GSEA.

2) Authors correctly used ComBat algorithm to eliminate batch effects and, for this reason, they should make explicit in supplementary figure B and C which samples belonged to the same dataset (using different symbols or colors) to highlight the dataset independence (technical variability) from the biological variability.

3) Authors should maintain the same color scheme: in figure 1B the legend bar (add a title for describing) shows positive magnitudo of fold change in red color, and blue for negative (and do the same also for high and low expression in survival analysis), while in the volcano plot (figure 1A) the color scheme is inverted. To avoid any doubt change consistently also the color of the circles in the Venn diagram (figure 1C). Explicit also the type of p-value correction in y-label in volcano plot.

7. PLOS authors have the option to publish the peer review history of their article (what does this mean?). If published, this will include your full peer review and any attached files.

Reviewer #1: No

Reviewer #3: No

---

## [Author Response · Author response to Decision Letter 1]

17 Nov 2021

Dear editor,

Thank you very much for your letter and advice. We have revised the paper, and would like to re-submit it for your consideration. We have addressed the comments raised by the reviewers, and the amendments are highlighted in red in Revised Manuscript with Track Changes. 

We hope that the revision is acceptable, and we look forward to hearing from you soon. 

With best wishes,

Yours sincerely,

Editor Comments:

Please review your reference list to ensure that it is complete and correct. Thanks for the comment. We have checked the reference list and made sure it is complete and correct.

Regarding the generation of prognostic nomograms, some clarifications and modifications are needed. Thank you for the reminder. Your suggestion is very meaningful. We have conducted a self-review based on the checklist according to your guidance. We have added and highlighted the missing descriptions in the manuscript, such as the patient population that generated the nomogram. We found that we met most of the requirements in the checklist, except for our study which may lack further validation. On the one hand, our study is one of the few that covers almost all common subtypes of soft tissue sarcoma and is therefore highly universal; however, on the contrary, validation of such a wide range of subtype models may not be so easy to achieve. In our study, we performed C-indexes and Calibration curves validation. Of course we clearly recognize that this is not sufficient and we have added a description of the shortcomings of the constructed nomogram in the discussion section, but we believe that our nomogram may be of some value as a guide and reference for subsequent related studies. Also, we fully understand your suggestion to remove the nomogram part, and we will comply with your suggestion to remove this section if you still feel that it is not necessary to keep it.

Reviewer #3 Comments:

We would like to express our sincere gratitude to the reviewer for the constructive and positive comments. It is an honor for our manuscript to receive such rigorous and meticulous guidance. Explain accordingly one by one:

1.Authors should specify which metric they used for ranking genes in GSEA. Thank you for your insightful suggestion. We have added a description of the GSEA ranking methodology in the results section.

2.They should make explicit in supplementary figure B and C which samples belonged to the same dataset. Thank you for your valuable comments. We have considered what you said before, but because the sample size included in our study was too large, adding 2 legends to one figure would have made the images very cluttered. After much testing and comparison, we finally chose the optimal approach and recreated the Figure S1. We divided the original image into 2 images and used different legend for each image.

3.Authors should maintain the same color scheme. Thanks for your reminder. We recreated the Figure 1 to ensure that the legend colors of the positive and negative expressions of the volcano plot, heat map, venn diagram, and survival curves remain consistent. We also clarified the description of the Y-label of the volcano plot. Also we added the missing legend to the heat map.

---

## [Editor Report · Decision Letter 2]

15 Dec 2021

PONE-D-21-24357R2Immunological and prognostic significance of novel ferroptosis-related genes in soft tissue sarcomaPLOS ONE

Dear Dr. Cui,

Thank you for submitting your manuscript to PLOS ONE. After careful consideration, we feel that it has merit but does not fully meet PLOS ONE’s publication criteria as it currently stands. Therefore, we invite you to submit a revised version of the manuscript that addresses the points raised during the review process.Please ensure that your decision is justified on PLOS ONE’s publication criteria and not, for example, on novelty or perceived impact.

We look forward to receiving your revised manuscript.

Kind regards,

Sandro Pasquali, M.D., Ph.D.

Academic Editor

PLOS ONE

Journal Requirements:

Additional Editor Comments:

Thank you for addressing reviewers and editor comments in your revised manuscript. Considering that nomograms are not validated, they should not be considered conclusive and therefore the statement in the conclusion of abstract and manuscript about nomogram should be removed. Also, can you specify whether nomograms were generated on primary or metastatic or both tumours and patients? This should clearly stated in the results section and discussion.

---

## [Author Response · Author response to Decision Letter 2]

18 Dec 2021

Dear editor,

Thank you very much for your letter and advice. We have revised the paper, and would like to re-submit it for your consideration. We have addressed the comments and the amendments are highlighted in red in Revised Manuscript with Track Changes. 

We hope that the revision is acceptable, and we look forward to hearing from you soon. 

With best wishes,

Yours sincerely,

Editor Comments:

- Please review your reference list to ensure that it is complete and correct. 

Thanks for the comment. We have checked the reference list and made sure it is complete and correct.

Additional Editor Comments:

- The statement in the conclusion of abstract and manuscript about nomogram should be removed.

Thank you for your suggestion. We have removed the statement of nomogram in the conclusion section.

- Can you specify whether nomograms were generated on primary or metastatic or both tumours and patients? This should be clearly stated in the results section and discussion.

Thank you for your reminder. We have clarified this issue in the result section and in the discussion section.

---

## [Editor Report · Decision Letter 3]

20 Dec 2021

Immunological and prognostic significance of novel ferroptosis-related genes in soft tissue sarcoma

PONE-D-21-24357R3

Dear Dr. Cui,

We’re pleased to inform you that your manuscript has been judged scientifically suitable for publication and will be formally accepted for publication once it meets all outstanding technical requirements.

Kind regards,

Sandro Pasquali, M.D., Ph.D.

Academic Editor

PLOS ONE

---

## [Editor Report · Acceptance letter]

24 Dec 2021

PONE-D-21-24357R3 

Immunological and prognostic significance of novel ferroptosis-related genes in soft tissue sarcoma 

Dear Dr. Cui:

I'm pleased to inform you that your manuscript has been deemed suitable for publication in PLOS ONE. Congratulations! Your manuscript is now with our production department. 

Kind regards, 

on behalf of

Dr. Sandro Pasquali 

Academic Editor

PLOS ONE